# The Antiaging Effect of Active Fractions and *Ent*-11α-Hydroxy-15-Oxo-Kaur-16-En-19-Oic Acid Isolated from *Adenostemma lavenia* (L.) O. Kuntze at the Cellular Level

**DOI:** 10.3390/antiox9080719

**Published:** 2020-08-08

**Authors:** Irmanida Batubara, Rika Indri Astuti, Muhammad Eka Prastya, Auliya Ilmiawati, Miwa Maeda, Mayu Suzuki, Akie Hamamoto, Hiroshi Takemori

**Affiliations:** 1Department of Chemistry, Faculty of Mathematics and Natural Sciences, IPB University, IPB Dramaga Campus, Bogor, West Java 16680, Indonesia; ilmiawati.aulia@gmail.com; 2Tropical Biopharmaca Research Center, IPB University, Taman Kencana Street No. 3, Bogor 16128, Indonesia; 3Department of Biology, Faculty of Mathematics and Natural Sciences, IPB University, IPB Dramaga Campus, Bogor, West Java 16680, Indonesia; rikaindriastuti@apps.ipb.ac.id (R.I.A.); muhammadprastyajuni2011@gmail.com (M.E.P.); 4Graduate School of Natural Science and Technology, Gifu University; Gifu 501-1193, Japan; Z4521078@edu.gifu-u.ac.jp; 5Department of Chemistry and Biomolecular Science, Faculty of Engineering, Gifu University; Gifu 501-1193, Japan; w3032081@edu.gifu-u.ac.jp (M.S.); ahama@gifu-u.ac.jp (A.H.)

**Keywords:** antiaging, *ent*-11α-hydroxy-15-oxo-kaur-16-en-19-oic acid, *Adenostemma lavenia*, *pap1*^+^ transcription factor, NRF2

## Abstract

Background: The extract of *Adenostemma lavenia* (L.) O. Kuntze leaves has anti-inflammatory activities and is used as a folk medicine to treat patients with hepatitis and pneumonia in China and Taiwan. The diterpenoid *ent*-11α-hydroxy-15-oxo-kaur-16-en-19-oic acid (11αOH-KA) is the major ingredient in the extract and has wide-spectrum biological activities, such as antitumor and antimelanogenic activities, as well as anti-inflammatory activity. However, the physical and biological properties of this compound as an antioxidant or antiaging agent have not been reported yet. Methods: In addition to in vitro assays, we monitored antioxidative and antiaging signals in *Schizosaccharomyces pombe* (yeast) and mouse melanoma B16F10 cells. Results: *A. lavenia* water and chloroform fractions showed antioxidant properties in vitro. The *A. lavenia* extracts and 11αOH-KA conferred resistance to H_2_O_2_ to *S. pombe* and B16F10 cells and extended the yeast lifespan in a concentration-dependent manner. These materials maintained the yeast mitochondrial activity, even in a high-glucose medium, and induced an antioxidant gene program, the transcriptional factor *pap1*^+^ and its downstream *ctt1*^+^. Accordingly, 11αOH-KA activated the antioxidative transcription factor NF-E2-related factor 2, NRF2, the mammalian ortholog of *pap1*^+^, in B16F10 cells, which was accompanied by enhanced hemeoxygenase expression levels. These results suggest that 11αOH-KA and *A. lavenia* extracts may protect yeast and mammalian cells from oxidative stress and aging. Finally, we hope that these materials could be helpful in treating COVID-19 patients, because *A. lavenia* extracts and NRF2 activators have been reported to alleviate the symptoms of pneumonia in model animals.

## 1. Introduction

*Adenostemma lavenia* (L.) O. Kuntze, a perennial herb, belongs to the Asteraceae family and grows widely in the tropical regions of Asia. This plant has been traditionally used as a medicinal herb in the Pacific Islands to cure pneumonia, lung congestion, fever, hepatitis, and skin wounds [1]. A recent study showed that *A. lavenia* extract had anti-inflammatory activity in LPS-treated RAW 264.7 cells and mice [2]. In addition, the leaf extract of this plant has antimelanogenic activities in B16F10 cells and suppresses hair pigmentation in infant mice [3,4]. Notably, *A. lavenia* leaf extract contains a higher amount of *ent*-11α-hydroxy-15-oxo-kaur-16-en-19-oic acid (11αOH-KA), which may be responsible for 50% of the antimelanogenic activity in the extracts [4] (Figure 1).

The 11αOH-KA belongs to diterpene/kaurane class, which is found in some plants, including the compositae (*Gochnatia decora*) and ferns (*Pteris semipinnata*), as well as *A. lavenia* [5,6]. Interestingly, 11αOH-KA exhibits various pharmaceutical potentials, such as anticancer, anti-inflammation, and skin whitening [4,5]. Despite the broad activities of 11αOH-KA, the potential of this compound as an antioxidant and antiaging material in skin care has not been clarified yet.

Aging is defined as a gradual loss of physiological integrity, which leads to the progressive deterioration of cellular components and constituents [7]. The accumulation of reactive oxygen species (ROS), mitochondrial dysfunctions, DNA mutations, and advanced glycation end products (AGEs) has been reported to be a primary factor in cellular aging [7,8]. However, aging phenotypes/symptoms are complicated in multicellular organisms due to the presence of different types of cells.

To examine the potential of 11αOH-KA as an antioxidant and antiaging reagent at cellular levels, we used *Schizosaccharomyces pombe* (yeast) and mouse B16F10 melanoma cells as model systems. *S. pombe* has a number of advantages including rapid growth, easy cultivation, and well-characterized genomes with conserved genetic pathways in eukaryotic cells, which facilitates its usage as a eukaryotic model to understand the cellular events that occur in higher organisms [9]. Excess calorie consumption and H_2_O_2_ stress have been found to modulate the lifespan in yeasts and other organisms with similar mechanisms [7,9,10,11,12].

*S. pombe* has mechanisms for controlling aging through the factors involved in oxidative stress responses [7,8]. In addition, calorie restriction (CR) conditions also extend the lifespan in *S. pombe* via the suppression of the Target of Rapamycin (TOR) pathway and upregulation of SIR2 histone deacetylase (sirtuin family) pathways. These cascades result in a suppressed mitochondrial activity, which enhances the expression of antioxidant enzymes and resistance to reactive oxygen species (ROS) [9,13]. In this pathway, the basic leucine zipper domain, bZIP, and transcription factor yeast Pap1 are crucial regulators of cellular defense against oxidative stresses [14].

On the other hand, in mammalian cells, antioxidative responses primarily activate the transcription factor nuclear factor E2-related factor 2 (NRF2, the yeast ortholog of Pap1). Then, the active NRF2 induces downstream genes including heme-oxygenase-1 (HO-1), *Nqo1*, and glutamate cysteine ligase catalytic (*Gclc*), which are essential for the inhibition of ROS-induced pro-inflammatory responses [15,16,17].

In this study, we demonstrated that *A. lavenia* water and chloroform fractions as well as 11αOH-KA prolonged yeast lifespan. Specifically, 11αOH-KA showed CR mimic activity in yeast, followed by oxidative stress responses. Furthermore, experiments in mouse B16F10 cells showed that these materials upregulate NRF2 protein levels, accompanied by enhanced levels of HO-1 protein. These results suggest beneficial effects of 11αOH-KA and *A. lavenia* extracts as potential candidates for antiaging ingredients in drugs, foods, supplements, and cosmetics.

## 2. Materials and Methods

### 2.1. 11α-OH-KA

*A. lavenia* was collected from Bogor, West Java, Indonesia. 11αOH-KA, isolated from *A. lavenia* leaves with a purity of >95%, judging from the NMR spectra in Gifu University [4], was dissolved in dimethyl sulfoxide (DMSO). Briefly, a dried powder (100 g) obtained from *A. lavenia* leaves was soaked in distilled water 1:30 (*w/v*) at 55 °C for 12 h. The water-soluble fraction (*A. lavenia* water fraction) was recovered after filtration with coffee filtrates, and 11αOH-KA was purified by chloroform extraction (*A. lavenia* chloroform fraction) followed by silica gel chromatography. 11αOH-KA was recovered as crystals.

### 2.2. Cell Culture

The fission yeast *S. pombe* wild-type strain ARC039 (h-leu1-32 ura4-294: Asahi Glass Co. Ltd., Tokyo, Japan), a gift from Dr. Hiroshi Takagi (Nara Institute of Science and Technology, Nara, Japan), was used in all experiments. The yeast cells were routinely maintained in a yeast extract with supplement (YES) medium containing 3% glucose.

Mouse B16F10 melanoma cells from the RIKEN cell bank (RIKEN, Saitama, Japan) were cultured in Dulbecco’s modified Eagle’s medium (DMEM with 4.5 g/L d-Glucose, Nacalai Tesque, Kyoto, Japan) containing 10% fetal bovine serum (FBS: Sigma-Aldrich Japan, Tokyo, Japan) and a penicillin‒streptomycin solution (WAKO, Osaka, Japan).

The B16F10 cells were plated in six-well dishes at 1.0 × 10^5^ cells/well. After two days, cells were washed with phosphate-buffered saline (PBS) and collected in a 1.5-mL tube. Melanin content was first visually examined by photos, and then measured by optical density (OD) at 450 nm after extraction with 2N NaOH for 4 h. The melanin content was normalized by protein levels measured by the Bradford method (Protein assay kit, Nacalai Tesque, Kyoto, Japan).

To measure NFR2 activity using a reporter assay, OKD48-luc plasmid (Transgenic Inc, Kobe, Japan) was transformed into B16F10 cells with pRL-TK (internal reporter, Promega, Madison, WI, USA), and the cells were stimulated with 11αOH-KA or andrographolide (AG) for 24 h. The reporter activities were monitored by the dual luciferase reporter system (Promega). To detect NRF2 protein, HO-1 protein, and Glyceraldehyde 3-phosphate dehydrogenase (GAPDH) protein, anti-NRF2 antibody (GTX103322, GeneTex Inc., Irvine, CA, USA), anti-HO-1 antibody (GTX101147, GeneTex Inc.), and horseradish peroxidase-conjugated anti-GAPDH antibody (MBL Co. Ltd., Nagoya, Japan) were used.

### 2.3. Measurement of Antioxidant and Antiglycation Activities

Antioxidant activities of fractions were measured based on the radical scavenging activity toward 2,2′-diphenyl-1-picrylhydrazyl (DPPH) and 2,2′-azino-bis-3-ethylbenzthiazoline-6-sulphonic acid (ABTS), as described previously [18,19]. The antiglycation reactions were carried out by mixing fractions in solution and the glycation substrate based on a previous method [20].

### 2.4. Oxidative Stress Tolerance and Survival Assays

*S. pombe* was cultured in a YES broth supplemented with fractions with an initial OD_600_ of 0.05 in a shaking incubator at 30 °C. The maximum concentration of fractions was set to 5 times IC_50_ in DPPH activity. At day 7 and 11, 5 mM H_2_O_2_ was added to the culture medium. The viability of *S. pombe* was measured at day 3 after H_2_O_2_ treatment. The survival assay was conducted by using the total plate count, TPC, method at day 11.

For the aging assay, spot tests were also conducted at day 7 and 11. Initially, each of the yeast cultures was adjusted to the OD_600_ of 1.0 and serially diluted. Immediately, 3 µL of each aliquot were spotted onto a YES or YES agar plates containing various concentrations of H_2_O_2_ and incubated for three days at 30 °C. As for chronological aging experiments, the TPC assay was performed on the yeast culture (similar with above) at day 1, 5, 10, 15, and 20. Each culture was serially diluted and spread in triplicate on a YES agar plate, followed by incubation for three days.

Yeast mitochondrial activity was determined by using rhodamine B (Merch, St. Louis, MO, USA) as a mitochondria probe. The reaction mixture was prepared as described in [11]. The fluorescent signal was observed using a BX51 fluorescent microscope (Olympus, Tokyo, Japan).

### 2.5. RNA Isolation and Real-Time Quantitative PCR Assay

Yeast cells were grown in a YES broth containing 11αOH-KA, with an initial OD_600_ of 0.05 at 30 °C for 18 h in a shaking incubator (120 rpm). Then, yeast cells were harvested with centrifugation at 5000 rpm for 2.5 min at −4 °C. RNA was isolated using RNAeasy kit (Qiagen, Germantown, MD, USA) and then converted to cDNA using iScriptTM cDNA Synthesis Kit (Bio-Rad, Hercules, CA, USA). Quantitative polymerase chain reaction (qPCR) was applied using Applied Biosystems StepOnePlus™ Instrument and Thunderbird SYBR qPCR master mix (Toyobo, Osaka, Japan) as a fluorescent mixture with the primers as follows: *pap1*^+^ Forward, 5′ TGGATGGCGATGTTAAGCCT/Reverse, 5′ GCAGCACGGTTTTGAGCTTT (SPAC1783.07c). *ctt1*^+^ Forward, 5′ TCGTGACGGCCCTATGAATG/Reverse, 5′ AGCAAGTGGTCGGATTGAGG (SPCC757.07c). Gene expression was normalized to the housekeeping gene act1. *act1*^+^ Forward, 5′ CGGTCGTGACTTGACTGACT/Reverse, 5′ ATTTCACGTTCGGCGGTAGT (SPBC32H8.12c).

### 2.6. Intracellular Yeast Metabolite Extraction and LC-MS Analysis

Intracellular yeast metabolites were prepared by the procedure described previously, with modifications [21]. Treatment cultures were prepared in YES broth medium with an initial yeast cells OD_600_ of 0.05 and supplemented with 11αOH-KA (45 µg mL^−1^), then incubated until the mid-log phase with constant shaking (120 rpm) at 30 °C. The harvested yeast cells were immediately quenched in 21 mL of MeOH at −20 °C. The extracellular metabolites were separated from intracellular metabolites (cells pellet) by centrifugation (5000 rpm) for 5 min at −20 °C. To extract metabolites, 2.5 mL pre-cold 50% MeOH/H_2_O was added to the yeast cells pellet, followed by supplementation in 2.5 mL of pre-cold CHCl_3_. After centrifugation (5000 rpm) for 5 min at −20 °C, the lower-phase (CHCl_3_) and the upper-phase (MeOH/H_2_O) were collected. This mixture was then concentrated by nitrogen evaporation and dry-frozen. Finally, each sample was resuspended in 100 µL (1:1, *v/v*) acetonitrile: H_2_O and 1 µL was used for each LC-MS injection.

LC-MS data were acquired using a UHPLC vanquish tandem equipped with UltimateTM 3000 RSLC nano system and coupled to an Q Executive hybrid quadrupole-orbitrap mass spectrometer (Thermo Fisher Scientific, Waltham, MA, USA). LC separation was conducted on an Accucore™ phenyl-hexyl HPLC column (Thermo Fisher Scientific, 100 × 2.1 mm, 2.6 µm particle size). Acetonitrile with 0.1% formic acid (A) and LC-MS H_2_O_2_ with 0.1% formic acid (B) were used as the mobile phase, with gradient elution from 95% A (5% B) to 5% A (95% B) in 30 min and a 0.3 mL min^−1^ flow rate. Electrospray ionization (ESI) was used. Each sample was injected once (1 µL) with the ESI, operated in both negative and positive ionization mode. Nitrogen was used as the carrier gas. The mass spectrometer was operated in full scan mode with a scan range of 100–1000 *m/z* and automatic data-dependent MS/MS fragmentation scans. Moreover, raw LC-MS data were analyzed by Compound Discoverer 2.1 software (Thermo Fisher Scientific). The corresponding software was integrated to the mzCloud and ChemSpider for matching fragmentation spectra and compounds.

### 2.7. Statistical Analyses

All results were expressed as the mean ± SEM (*n* = 3). Means of different groups were compared using one-way ANOVA followed by Duncan’s multiple range test.

## 3. Results

### 3.1. Ability of A. lavenia Fractions to Scavenge Free Radicals and Inhibit AGEs Production In Vitro

Both the *A. lavenia* chloroform fraction (*Acf*) and water fraction (*Awf*) could substantially scavenge radicals of DPPH (Table 1). However, their scavenging efficacies were about fifty times lower than that of ascorbic acid. In contrast, based on an ABTS scavenging assay (represented by the value equivalent to Trolox), both fractions efficiently scavenged radicals. An antiglycation assay showed that *Acf* has the higher capacity to suppress AGEs production than *Awf*. These results suggest that *A. lavenia* extracts not only had radical-scavenging activity, but also antiglycation activity, which prompted us to examine the antiaging potential of the *A. lavenia* fractions and its ingredient 11αOH-KA in a model organism, *S. pombe*.

### 3.2. Cellular Antioxidant and Antiaging Activities of A. lavenia Extracts

Oxidative stress is a major factor of aging [22], and a number of natural compounds have been found to have antiaging activity. Therefore, we examined the correlations between the antioxidative activities and antiaging potentials of *A. lavenia* fractions. First, we confirmed no effect of DMSO on yeast survival under oxidative stress conditions with 5 mM H_2_O_2_ (Figure 2A).

When *Awf* and *Acf*, at different doses, were supplemented to the medium, dose-dependent protection of *S. pombe* from 5 mM H_2_O_2_ was observed (Figure 2B). The limiting dilution indicated optimum protective concentrations of *Awf* and *Acf* at 1260 µg mL^−1^ and 888 µg mL^−1^, respectively (Figure 2C). The calculation of colony-forming units suggested that *Acf* was more effective than *Awf* (Figure 2D).

### 3.3. 11αOH-KA Extends Yeast Life Span

To examine whether 11αOH-KA, a major ingredient in *A. lavenia* fractions [4], conferred H_2_O_2_ resistance and longevity to *S. pombe*, we performed similar experiments and a chronological life span (CLS) assay with 11αOH-KA. In spot assays, the 11αOH-KA treatment (45 µg mL^−1^) significantly extended the yeast lifespan at both day 7 and 11 (Figure 3A). In addition, although the effect on the longevity under calorie restriction (CR: 0.5% glucose) conditions was higher than the 11αOH-KA treatment, the presence of 45 µg mL^−1^ 11αOH-KA substantially prolonged the yeast survival under normal calorie conditions (3% glucose) (Figure 3B). Furthermore, 11αOH-KA significantly improved the cell growth of *S. pombe* at day 11, when mild oxidative stresses were loaded by 3 mM H_2_O_2_ (Figure 3C). Notably, a protective effect of 11αOH-KA against H_2_O_2_ was also observed under the CR conditions.

### 3.4. A. lavenia-Derived Fractions and 11αOH-KA Treatment Increase Mitochondria Activity

It is known that CR conditions enhance mitochondrial activity. Indeed, CR treatment (0.5% glucose) provoked high mitochondrial activity in yeast (Figure 4A). Interestingly, treatment of *Awf* (1260 µg mL^−1^), *Acf* (888 µg mL^−1^), and 11αOH-KA (45 µg mL^−1^) enhanced mitochondrial activity in the 3% glucose medium (Figure 4B), suggesting that *A. lavenia* treatment might mimic CR conditions.

### 3.5. 11αOH-KA Treatment Alters Yeast Intracellular Metabolites Involved in Stress Response Mechanism

Based on metabolomics profile, about 85 metabolites (Appendix A, Appendix A) could be analyzed using the software Compound Discoverer 2.1, and we observed a fluctuation in nearly all 83 metabolites (97.6%: 30 increased and 53 decreased) in *S. pombe* treated with 11αOH-KA. Notably, the levels of 15 metabolites were significantly changed after supplementation with 11αOH-KA (Table 2). The decrease in lactic acid level and the preservation of glucose in the yeast cells treated with 11αOH-KA might result from the activation of mitochondrial functions. Surprisingly, we found that two metabolites, l-proline and l-arginine (stress protectants), decreased significantly—by 33.75- and 2.25-fold, respectively. On the contrary, betaine and choline (other stress protectants) were significantly increased by 7.5- and 1.85-fold, respectively, suggesting a cellular homeostasis alteration occurred following treatment with 11αOH-KA that involves a wide array of stress response mechanisms, which might lead to the lifespan extension of *S. pombe*.

Importantly, there was no increase in l-cysteine, which is important for the synthesis of the antioxidant glutathione, suggesting that the Pap1 transcription factor, the yeast equivalent of mammalian NRF2 [23], might not fully contribute to the 11αOH-KA-mediated antioxidative stress actions in *S. pombe*.

### 3.6. A. lavenia Fractions and 11αOH-KA Upregulate Pap1-Dependent Antioxidant Signaling in Yeast

Although less of a contribution of the transcription factor Pap1 to the antioxidant signaling induced by 11αOH-KA was expected, we examined the effect of *A. lavenia* fractions and 11αOH-KA on Pap1 mRNA expression. Unexpectedly, treatment with all materials (*Awf*, *Acf*, and 11αOH-KA) significantly upregulated the expression of the *pap1* gene, while only *Acf* and 11αOH-KA induced the *ctt1* gene expression, a downstream product of Pap1 (Figure 5A,B), suggesting that unknown factors in *Awf* might suppress the expression of the *ctt1* gene.

### 3.7. 11αOH-KA Upregulates Antioxidant Signaling in Mouse B16F10 Melanoma Cells

NRF2 is the orthologue of Pap1 in mammals [24,25], and several kauranic acids as well as AG (andrographolide: other diteropene) have been reported to decrease the rate of melanogenesis and increase the levels of the antioxidant enzyme HO-1 via NRF2 [26,27]. Therefore, we examined the possible involvement of NRF2 in 11αOH-KA-mediated antimelanogenic action and antioxidative stress pathways. Both 11αOH-KA and AG suppressed melanogenesis in mouse melanoma B16F10 cells (Figure 6A,B). The efficacity of 11αOH-KA was higher than AG.

To monitor Nrf2 activity in B16F10 cells, we used the NRF2-responsible reporter (OKD48-luc) (Figure 6C). 11αOH-KA strongly upregulated NRF2 activity, while AG did only slightly. We could detect free NRF2 protein when the cells were treated with 10 µM 11αOH-KA, which was accompanied by the induction of HO-1 (Figure 6D). Although *Acf* weakly upregulated protein levels of Nrf2 and HO-1, *Awf* did significantly (Figure 6E), suggesting that unidentified ingredients might negatively or positively interact with 11αOH-KA. In addition, 11αOH-KA and *A. lavenia* fractions conferred resistance to H_2_O_2_ on B16F10 cells (Figure 6F,G), suggesting that 11αOH-KA might induce antioxidative signaling, and these cascades might be implicated in antimelanogenic activity in mouse melanocytes.

## 4. Discussion

A water-based extract of *A. lavenia* leaves has been used for the treatment of inflammation, pneumonia, fever, hepatitis, lung congestion, and digestive system disorders [1,2,3,28], and, in the present study, exhibits additional functions, such as antioxidant and antiglycation activities. In addition to these benefits, we have found that *A. lavenia* fractions and 11αOH-KA promote longevity in *S. pombe* and resistance to oxidative stress in *S. pombe* as well as mouse B16F10 cells.

The *A. lavenia* leaf extract contained a high amount of 11αOH-KA (approximately 2.5% of dry leaf weight), and the compound showed antimelanogenic activity [4]. The in vitro antioxidant and antiglycation capacities of *Awf* and *Acf* were relatively weak compared to positive controls [3]. However, the fractions may have stronger antioxidant and antiglycation activities than other Asteraceae extracts. For example, extracts from *Erigeron caucasicus* and *Faujasiopsis flexuosa* have an IC_50_ value of 704 µg mL^−1^ and glycation inhibition value of 10.23% from a 1000 µg mL^−1^ sample for DPPH and antiglycation activities, respectively, which is approximately three times less effective than *A. lavenia* [29,30].

Although *A. lavenia* has not been approved for medical use, QualiHerb Co. Ltd. produces a water extract of aerial parts of *A. lavenia* in Taiwan and the United States. The supplier recommends taking the extract (0.4–1.2 g) two or three times a day before meals. When we imported the extract and reconstituted it in water (30 folds), it contained 11αOH-KA with only 1/100 of our water extracts (<10 µg mL^−1^) [4], almost 5-fold less concentration (even without further dilution) than the optimal concentration of 11αOH-KA in the present experiment.

*S. pombe* is commonly used as a model organism in aging studies [31]. Similar to several natural compounds, including acivicin, tschimganine, and l-arginine, it has been shown to extend the lifespan [11,12,22,31,32], in cooperation with the effects of CR [7]. *A. lavenia* fractions (*Acf* and *Awf*) and 11αOH-KA showed longevity effects as well as resistance to H_2_O_2_ oxidative stress in yeast. The longevity effects have been reported to be mediated by downregulation of the nutrient-sensing pathways involved in Tor1, Sck2, and Pka1 [31]. These factors modulate other cellular factors and events, such as Sir2, autophagy, and the adaptive responses, which, coupled with the downregulation of mitochondrial activities, lead to resistance to oxidative stress.

Interestingly, we observed that *A. lavenia* fractions and 11αOH-KA could upregulate yeast mitochondrial activity, which was also supported by metabolomics analyses. Mitochondria are indispensable in all eukaryotes to generate the bulk of cellular ATP and provide intermediates of amino acids, nucleotides, and lipids [33]. Importantly, mitochondrial activity produces intracellular ROS as by-products and these molecules play a critical role in regulating the yeast lifespan [34]. If the ROS level exceeds a toxic threshold, it accelerates the aging process by eliciting oxidative damage in yeast cells.

In contrast, if the concentration of ROS is maintained at a hormetic level (i.e., insufficient to cause damage to cellular macromolecules), the ROS can activate signaling networks that further induce gene expression for adaptive responses [15,17]. In agreement with those theories, we suggest that *A. lavenia* fractions and 11αOH-KA as well as CR may enforce mitochondrial integrity, resulting in a low level of ROS production. A similar mechanism has already been reported for compounds such as 3.3-diindolemethane in an extract of *Pseudomonas* sp. that promotes yeast longevity through ROS-adaptive signaling, which modulates mitochondrial activity [12,35]. In the future, we have to monitor the ROS production in cells treated with 11αOH-KA and *A. lavenia* extracts.

In addition, *A. lavenia* fractions stimulated the expression level of some genes implicated in oxidative stress responses, including *pap1*^+^ and *ctt1*^+^. The transcription factor Pap1 is mainly involved in adaptation rather than survival responses. Once Pap1 is activated and transported into the nucleus, it induces the expression of the following oxidative stress-induced genes, among others: *ctt1*^+^, *trx2^+^*, *trr1^+^*, and *pgr1^+^* [17]. The products of these genes are involved in scavenging ROS, recovery from cell damage, and adaptive stress responses.

Similar to Pap1 in yeast cells, its orthologue factor NRF2 may also be activated by 11αOH-KA in B16F10 cells. It has already been reported that the ethyl acetate fraction of *A. lavenia* promotes the NRF2‒HO-1 axis and protects the lungs from lipopolysaccharide-induced inflammation [2]. However, this report proposed that *p-*coumaric acid was the compound responsible for the anti-inflammatory activity, suggesting that several compounds in *A. lavenia* may contribute to the antioxidative and anti-aging activities. Although NRF2 is known to suppress melanogenesis via the downregulation of *Mitf* (microphthalmia-associated transcription factor) gene expression, 11αOH-KA does not lower the MITF expression level [4]. This evidence suggests that NRF2 may contribute to the antimelanogenic activity of 11αOH-KA and *A. lavenia* extracts.

To gain comprehensive insight into 11αOH-KA’s effects on the extension of the yeast lifespan, we analyzed the metabolomics profile. Interestingly, we found that 11αOH-KA treatment remarkably decreased l-proline and l-arginine metabolites. l-proline and l-arginine play a pivotal role in yeast cells’ resistance to various stresses, i.e., against freezing, desiccation, oxidation, and ethanol [36,37]. In association with the metabolic pathway, the biosynthesis of l-arginine involves l-proline as one of the substrates. Along with those reports, the overexpression of genes that substantially promote l-proline biosynthesis increases the intracellular nitric oxide (NO) level in *S. cerevisiae*. As a result, increased NO confers high tolerance to multiple stresses including oxidation, drying, and freezing [38,39], as well as antiaging activity. However, in mouse macrophages, kaurenoic acids have been found to suppress NO production [40]. Further study is required to measure the NO level due to 11αOH-KA treatment.

In a yeast metabolic map, choline is one of the substrates for betaine biosynthesis via the betaine‒aldehyde pathway. It is intriguing to note that betaine has a principle role in the acquisition of stress tolerance in various organisms, including animals, plants, and most microorganisms, against environmental stresses such as drought, oxidative stress, osmotic stress, and extreme temperatures [41]. In fact, betaine also plays an important role in some yeast cells, including *S. cerevisiae* and *Candida oleophila*, inducing osmotic and oxidative stresses [42,43]. Moreover, recent studies have shown that betaine supplementation could have an antioxidative effect in HepG2 cells and rat brains by reducing intracellular levels [44,45]. This raises the possibility that the high level of betaine in yeast cells might function as an osmoprotectant that further induces yeast cells to adapt to stressful environments and, thus, increases the lifespan.

In addition to its antioxidant properties, 11αOH-KA has antimelanogenic potential, which can provide additional knowledge for the development of drugs, food supplements, or antiaging cosmetics. In particular, the activation of the NRF2‒HO-1 axis is now proposed to be a candidate to treat COVID-19 patients [46,47,48], and *A. lavenia* extract has been shown to ameliorate the pathogenesis of a model pneumonia induced by lipopolysaccharides [2]. We hope that the present study gives clues that will help us to solve a wide range of problems in the future.

## 5. Conclusions

11αOH-KA has unique physical properties as an antioxidant and induces cellular factors (*pap1*/*ctt1* and NRF2/HO-1 in *S. pombe* and mouse melanoma, respectively) that contribute to resistance to oxidative stress. Specifically, 11αOH-KA extends the lifespan of *S. pombe* cells and protects both yeast and mouse cells from H_2_O_2_. These results suggest that 11αOH-KA and its source, *A. lavenia*, can be attractive materials for antiaging and related diseases.

## Figures and Tables

**Figure 1 antioxidants-09-00719-f001:**
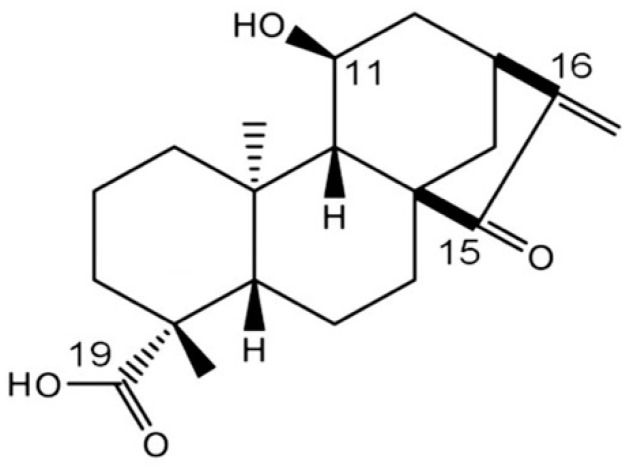
The structure of *ent*-11α-hydroxy-15-oxo-kaur-16-en-19-oic acid (11αOH-KA).

**Figure 2 antioxidants-09-00719-f002:**
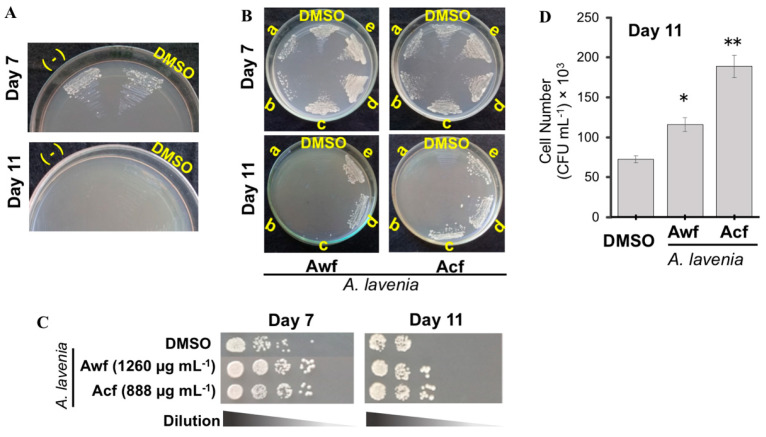
Antioxidative activity of *A. lavenia* extracts. (**A**) *S. pombe* was preincubated in the YES liquid medium with or without (−) DMSO for seven or 11 days and then streaked on agar plates containing 5 mM H_2_O_2_. (**B**) The effects of *A. lavenia* fractions on the oxidative stress resistance in the yeast cells were examined by the streaking method. *A. lavenia Awf* (a, 252; b, 504; c, 756; d, 1008; e, 1260 µg mL^−1^) or *Acf* (a, 222; b, 444; c, 666; d, 888; e, 1110 µg mL^−1^) were mixed in the YES liquid medium. At day 7 or 11, the yeast cells were streaked on plates supplemented with 5 mM H_2_O_2_. (**C**) The yeast cells that had been preincubated as in (**B**) were diluted and spotted on agar plates. (**D**) The number of CFU (colony-forming units) was obtained after 11 days of incubation of (**C**). Error bars represented S.D. from experiments performed in triplicate (*n* = 5, * *p* < 0.05, ** *p* < 0.01).

**Figure 3 antioxidants-09-00719-f003:**
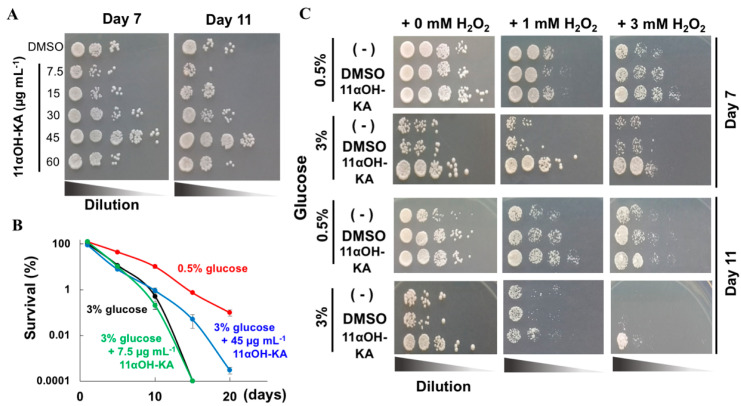
11αOH-KA promotes longevity in yeast. (**A**) Spot assay of *S. pombe* with various concentrations of 11αOH-KA. (**B**) Colony-forming units (CFU) were measured after the incubation of the yeast with 11αOH-KA. Data are normalized by the values of DMSO control at day 1 (stationary phase). The y-axis is shown in logarithmic scale. Yeasts were cultured in YES liquid medium containing either 0.5% glucose with DMSO (red) or 3% glucose with DMSO (black), 11αOH-KA (45 µg mL^−1^: blue), or 11αOH-KA (7.5 µg mL^−1^: green) *n* = 3. (**C**) The yeast cells were preincubated in YES liquid medium containing DMSO or 11αOH-KA treatment (45 µg mL^−1^) with glucose (0.5% or 3%). Mild oxidative stress was induced by the addition of H_2_O_2_ (1 mM or 3 mM).

**Figure 4 antioxidants-09-00719-f004:**
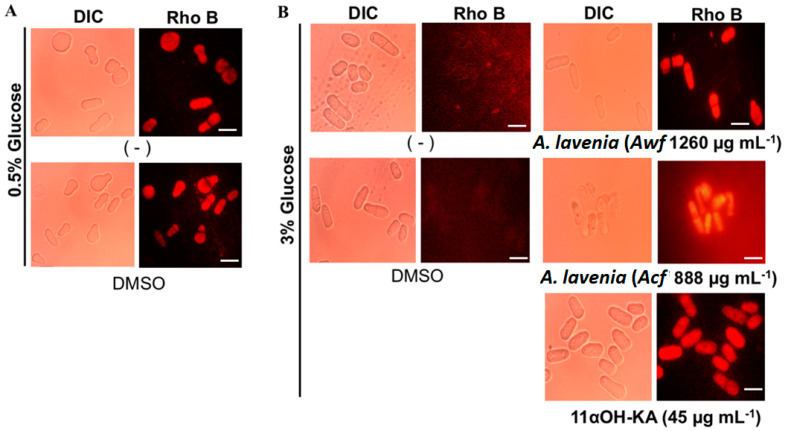
Effect of *A. lavenia* fractions and 11αOH-KA on mitochondria activity. (**A**) *S. pombe* was incubated under CR conditions (0.5% glucose), and mitochondria with a high membrane potential were stained with rhodamine B (Rho B: red fluorescence signals). (**B**) The yeast cells were treated with fractions of *A. lavenia* water (*Awf*) or chloroform (*Acf*) or 11αOH-KA and stained with Rho B. DIC: Differential interference contrast. Bars represent 5 µm.

**Figure 5 antioxidants-09-00719-f005:**
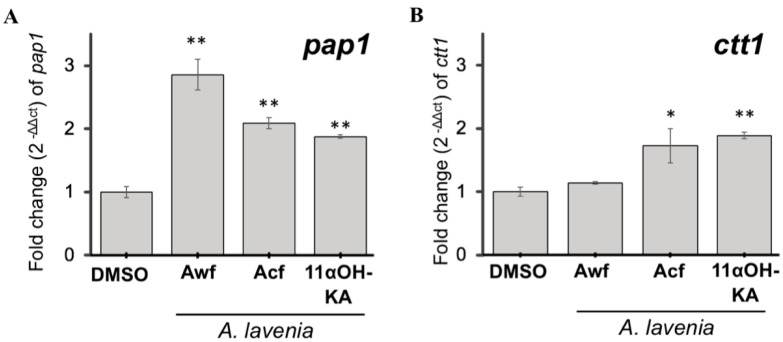
*A. lavenia* fractions and 11αOH-KA induced yeast antioxidative stress pathways. The yeast cells were cultured in YES liquid (3% glucose) supplemented with *A. lavenia* water (*Awf*: 1260 µg mL^−1^), chloroform *Acf*: (888 µg mL^−1^) fractions, and 11αOH-KA (45 µg mL^−1^) for 18 h. The expression levels of the *pap1*^+^ (**A**) and *ctt1*^+^ (**B**) genes were measured by quantitative PCR analysis. Values were normalized with those in the DMSO group. Means and S.D. are shown. (*n* = 3) *: *p* < 0.05, **: *p* < 0.01.

**Figure 6 antioxidants-09-00719-f006:**
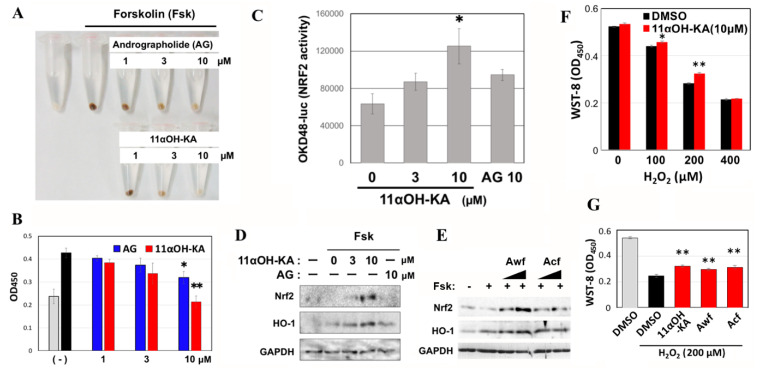
11αOH-KA upregulates Nrf2 pathways. B16F10 cells were treated with the indicated concentrations of 11αOH-KA or andrographolide (AG) for 24 h. The cells were recovered into sample tubes, and melanin contents were evaluated by photos (**A**) and optical density after lysis with NaOH (**B**). (**C**) B16F10 cells that had been transformed with the Nrf2 reporter (OKD48-luc) were treated with 11αOH-KA or AG for 24 h. Means and S.D. are shown (*n* = 3). *: *p* < 0.05. (**D**) The cells were lysed with SDS sample buffer without the reducing agent mercaptoethanol and used for Western blot analyses. (**E**) *Aw**f* (400 or 1200 µg mL^−1^) and *Acf* (300 or 900 µg mL^−1^) were used as stimulants. (**F**) B16F10 cells that had been precultured with or without 10 µM of 11αOH-KA for 24 h were treated with H_2_O_2_ for a further 24 h. Cell variability was examined by the Cell Counting kit (WST-8). (*n* = 4) ** *p* < 0.01. (**G**) The same experiments were performed with 11αOH-KA (10 µM), *Awf* (1260 µg mL^−1^), and *Acf*: (888 µg mL^−1^) in the presence of 200 μM H_2_O_2_. Gray and black bars indicate without H_2_O_2_ treatment and with only H_2_O_2_ treatment, respectively.

**Table 1 antioxidants-09-00719-t001:** In vitro antioxidant and antiglycation activities of *A. lavenia* fractions.

Samples	Fractions	Antioxidant Activities		Antiglycation Activity (% Inhibition of 1000 µg mL^−1^ Sample)
		DPPH IC_50_ (µg mL^−1^)	ABTS (mg Trolox/g Sample)	
*A. lavenia*	H_2_O (*Awf*)	252.02 ± 3.23 *	3.63 ± 0.41	8.87 ± 2.28 *
	CHCl_3_ (*Acf*)	222.37 ± 1.16 *	3.24 ± 0.39	33.44 ± 4.87 *
Ascorbic acid		4.06 ± 0.03	-	-
Aminoguanidine		-	-	73.00 ± 3.26

IC_50_ indicates the ability of fractions to scavenge DPPH free radicals. The values are means and standard deviations. *A. lavenia* water fraction (*Awf*) and chloroform fraction (*Acf*) contain 11αOH-KA at 15.1% and 56.6% (*w/w*), respectively. Statistically significant differences in the same column were determined by one-way ANOVA followed by Duncan’s multiple range test (*: significantly different from positive controls, *p* < 0.05). -: not detectable.

**Table 2 antioxidants-09-00719-t002:** Analyses intracellular yeast metabolites following 11αOH-KA (45 µg mL^−1^) treatment, as analyzed by the LC/MS approach.

Metabolites	Abundance (%)	
	DMSO	11αOH-KA
l-Methionine	0.10 ± 0.01	0.07 ± 0.01 *
l-Proline	1.35 ± 0.04	0.04 ± 0.01 *
Leucine	0.50 ± 0.03	0.53 ± 0.02
l-Phenylalanine	0.11 ± 0.02	0.35 ± 0.06 *
l-Serine	0.17 ± 0.01	0.27 ± 0.01 *
l-Tyrosine	0.13 ± 0.01	0.20 ± 0.01 *
Betaine	0.66 ± 0.20	4.96 ± 0.32 *
l-Arginine	0.18 ± 0.01	0.08 ± 0.01 *
l-Glutamic acid	0.21 ± 0.02	0.32 ± 0.02 *
1-Methylguanosine	0.55 ± 0.01	0.32 ± 0.03 *
l-Lactic acid	0.22 ± 0.01	0.08 ± 0.01 *
d-Glucose	0.08 ± 0.01	0.65 ± 0.03 *
Choline	0.07 ± 0.01	0.13 ± 0.01 *
Biotin	4.07 ± 0.22	4.33 ± 0.62
Nicotinamide	0.17 ± 0.01	0.09 ± 0.01 *

Statistical analyses were performed by one-way ANOVA followed by multiple Duncan range test analysis (*n* = 3). (*: significantly different from DMSO control treatment *p* < 0.05).

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
