# Peer review of "The Antiaging Effect of Active Fractions and Ent-11α-Hydroxy-15-Oxo-Kaur-16-En-19-Oic Acid Isolated from Adenostemma lavenia (L.) O. Kuntze at the Cellular Level"

_antioxidants, 2020, doi:10.3390/antiox9080719_

Round 1

Reviewer 1 Report

In the MS it has been demonstrated that water fractions of A. Lavenia could be have anti-aging properties. 11αOH-KA showed CR mimic activity in yeast. The paper is a following work from other paper from the same authors Foods 2020 Jan 9;9(1):73, doi: 10.3390/foods9010073. The paper is well written and could be published. Some suggestions:

the conclusions may be more comprehensive

Author Response

Thank you very much for your suggestion. We rewrote the conclusion as :

11αOH-KA has unique physical properties as an antioxidant and induces cellular factors (Pap1/Ctt1 and NRF2/HO-1 in S. pombe and mouse melanoma, respectively) that contribute to resistance to oxidative stress. Specifically, 11αOH-KA extends the lifespan of S. pombe cells and protects both yeast and mouse cells from H2O2. These results suggest that 11αOH-KA and its source, A. lavenia, can be attractive materials for antiaging and related diseases. 

Reviewer 2 Report

This research article is aimed at evaluating the prospective antioxidant/antiaging effect of 11αOH-KA isolated from Adenostemma lavenia in yeast and mouse melanoma cells. The main results demonstrated that Adenostemma lavenia leaf extracts confer resistance to oxidative stress in both the experimental models. Importantly, the effects of these extracts were also recapitulated by the administration of 11αOH-KA, suggesting that this compound is the most important molecule in mediating the outcomes induced by Adenostemma lavenia. At molecular level, Adenostemma lavenia-derived fractions and 11αOH-KA increase mitochondria activity in yeast, and significantly induces the activation of the antioxidant response mediated by Pap1/NRF2.

When evaluated as a whole, the manuscript appears interesting and the experimental design is adequate. However, there are few shortcomings that should be addressed:

1) English language should be carefully revised by a native english speaker, as several syntax and grammar errors are present in the whole text.

2) This work would gain much more strength if experiments related to figure 6 will be accompanied by evaluating 11αOH-KA and A. lavenia extracts on direct/indirect measurements of ROS. Indeed, the expression of NFR2/HO1 pathway should be associated by a concurrent analysis of cellular ROS content. DCF assay or indirect assessment of oxidative nucleid acid damage (immunofluorescence for 8-OHdG) could be performed. In addition, NRF2/HO1 pathway should be evaluated by Western blot not only in 11αOH-KA-treated cells, bu also in B16F10 stimulated with A. lavenia extracts.

Author Response

1) English language should be carefully revised by a native english speaker, as several syntax and grammar errors are present in the whole text.

Thank you very much for your suggestion. We asked a native English speaker (via MDPI system) to prove our manuscript.

2) This work would gain much more strength if experiments related to figure 6 will be accompanied by evaluating 11αOH-KA and A. lavenia extracts on direct/indirect measurements of ROS. Indeed, the expression of NFR2/HO1 pathway should be associated by a concurrent analysis of cellular ROS content. DCF assay or indirect assessment of oxidative nucleid acid damage (immunofluorescence for 8-OHdG) could be performed. In addition, NRF2/HO1 pathway should be evaluated by Western blot not only in 11αOH-KA-treated cells, bu also in B16F10 stimulated with A. lavenia extracts.

Thank you for your suggestion. we examined the levels of NFR2/HO1 protein in B16F10 cells (Fig. 6E). Both the water (Awf) and chloroform fractions (Acf) upregulated the protein levels of NRF2/HO-1. However, the results did not show a completely correlation with those in yeast cells. Some unknown ingredients might moderate the actions of 11αOH-KA. This was mentioned in Results as :

Although Acf weakly upregulated protein levels of Nrf2 and HO-1, Awf did significantly (Fig. 6E), suggesting that unidentified ingredients might negatively or positively interact with 11αOH-KA.

We could not obtain a kit to measure 8-OHdG due to disrupted research environments (import systems) caused by COVID-19.   

Reviewer 3 Report

The authors describe findings suggestive of anti-oxidant, anti-aging actions of Adenostemma lavenia (L.) O. Kuntze in in-vitro studies. The methods appear appropriate for investigating these effects in yeast and mouse cells and the results are thoroughly explained. However, the manuscript contains numerous errors in English (singular/plural, use of articles, sentence structure, spelling, and tense). Examples of lines containing errors:

26, 27, 31, 35, 47, 49, 57, 76, 89, 97, 136, 202, 213, 225, 270, 273, 287, 299, 308, 321, 331, 357, 359, 361, 378, 382 (reducing levels of what?), 382 (these).

From what company/entity was S. Pombe obtained for the experiments?

Table 2: include significance levels

Discuss the translation of the concentrations of A. lavenia used in the experiments to concentrations appropriate for human consumption.

Lines 259, 260 - provide citations for these properties (stress protectant; oxidative stress inducers) as you do in the discussion. Also, provide citations for lines 263-5; 282-4.

Discussion: include strengths and limitations.

Conclusion: needs improvement.

Author Response

The authors describe findings suggestive of anti-oxidant, anti-aging actions of Adenostemma lavenia (L.) O. Kuntze in in-vitro studies. The methods appear appropriate for investigating these effects in yeast and mouse cells and the results are thoroughly explained. However, the manuscript contains numerous errors in English (singular/plural, use of articles, sentence structure, spelling, and tense). Examples of lines containing errors:

26, 27, 31, 35, 47, 49, 57, 76, 89, 97, 136, 202, 213, 225, 270, 273, 287, 299, 308, 321, 331, 357, 359, 361, 378, 382 (reducing levels of what?), 382 (these).

Thank you very much for your suggestion. We asked a native English speaker (via MDPI system) to proof our manuscript.

382 : the reducing agent mercaptoethanol

From what company/entity was S. Pombe obtained for the experiments?

Thank you for your suggestion: We mentioned the origin of the yeast.

The fission yeast S. pombe wild-type strain ARC039 (h-leu1-32 ura4-294: Asahi Glass Co. Ltd., Tokyo, Japan.), a gift from Dr. Hiroshi Takagi (NAIST, Japan), was used in all experiments.

Table 2: include significance levels

We confirmed reproducibility and added mark of significance. All metabolites showed significant changes.  

Statistically significant differences in the same metabolites were performed by one-way ANOVA followed by multiple Duncan range test analysis (n=3). (*: significantly different from DMSO control treatment p < 0.05).

Discuss the translation of the concentrations of A. lavenia used in the experiments to concentrations appropriate for human consumption.

Thank you for your suggestion. We have no data about human consumption. However, we can obtain a commercial product (A. lavenia extracts) as a supplements. We added information of these product in Disucussion.

Although A. lavenia has not been approved for medical use, QualiHerb Co. Ltd. produces a water extract of aerial parts of A. lavenia in Taiwan and the United States. The supplier recommends taking the extract (0.4‒1.2 g) two or three times a day before meals. When we imported the extract and reconstituted it in water (30 folds), it contained 11αOH-KA with only 1/100 of our water extracts (<10 µg mL-1) [4], almost 5-fold less concentration (even without further dilution) than the optimal concentration of 11αOH-KA in the present experiment.

Lines 259, 260 - provide citations for these properties (stress protectant; oxidative stress inducers) as you do in the discussion. Also, provide citations for lines 263-5; 282-4.

Thank you for your indications

  1. Kim, S.J.; Kim, H.G.; Kim, B.C.; Park, E.H.; Lim, C.J. Transcriptional regulation of glutathione synthetase in the fission yeast Schizosaccharomyces pombe. Mol. Cells. 2004, 18, 242-248.
  2. Simaan, H.; Lev, S.; Horwitz, B.A. Oxidant-sensing pathways in the responses of fungal pathogens to chemical stress signals. Front. Microbiol. 2019, 10, 567, doi: 10.3389/fmicb.2019.00567.
  3. D'Autréaux, B.; Toledano, M.B. ROS as signalling molecules: mechanisms that generate specificity in ROS homeostasis. Nat. Rev. Mol. Cell. Biol. 2007, 8, 813-824, doi: 10.1038/nrm2256.
  4. Zhu P.Y.; Yin W.H.; Wang M.R.; Dang Y.Y.; Ye X.Y. Andrographolide suppresses melanin synthesis through Akt/GSK3β/β-catenin signal pathway. J. Dermatol. Sci. 2015, 79, 74–83, doi: 10.1016/j.jdermsci.2015.03.013.
  5. Mussard, E.; Cesaro, A.; Lespessailles, E.; Legrain, B.; Berteina-Raboin, S.; Toumi, H. Andrographolide, a Natural Antioxidant: An Update. Antioxidants. 2019, 8, 571, doi: 10.3390/antiox8120571.

Discussion: include strengths and limitations.

Thank you for your suggestion. We have no precise information. We cited only mouse models. (refence No. 2). Please see also the response below.

Conclusion: needs improvement.

Thank you for your suggestion. At first, I added a sentence

“Finally, we hope that these materials could be helpful in treating COVID-19 patients, because A. lavenia extracts and NRF2 activators have been reported to alleviate the symptoms of pneumonia in model animals.   “  in abstract.

In Disucssion

In particular, the activation of the NRF2‒HO-1 axis is now proposed to be a candidate to treat COVID-19 patients [46‒48], and A. lavenia extract has been shown to ameliorate the pathogenesis of a model pneumonia induced by lipopolysaccharides [2]. We hope that the present study gives clues that will help us to solve a wide range of problems in the future.  

  1. Horowitz, R.I; Freeman, P.R. Three novel prevention, diagnostic, and treatment options for COVID-19 urgently necessitating controlled randomized trials. Med. Hypotheses. 2020 143, 109851. doi: 10.1016/j.mehy.2020.109851.
  2. McCord, J.M.; Hybertson, B.M.; Cota-Gomez, A.; Geraci, K.P.; Gao, B. Nrf2 Activator PB125 ® as a Potential Therapeutic Agent against COVID-19. Antioxidants (Basel). 2020 9, 518. doi: 10.3390/antiox9060518.
  3. Cuadrado. A.; Pajares, M.; Benito, C.; Jiménez-Villegas, J; Escoll, M.; Fernández-Ginés, R.; Yagüe, A.J.G.; Lastra, D.; Manda, G.; Rojo, A.I.; Dinkova-Kostova, A.T. Can Activation of NRF2 Be a Strategy against COVID-19? Trends in Pharmacological Sciences 2020 in press

This is just speculation, not “strengths and limitations”.

If you feel these sentences are too speculative, we will remove them. 

Round 2

Reviewer 1 Report

The paper could be published in the present form.

Reviewer 2 Report

The authors provided a clear point-by-point response letter, and addressed most of the reviewer's comments. I still continue to believe that direct/indirect measurements of ROS would strengthen the conclusions of an experimental work mainly focused on antioxidant pathways, however I understand the technical problems caused by COVID19.

Author Response

(Comment)

 I still continue to believe that direct/indirect measurements of ROS would strengthen the conclusions of an experimental work mainly focused on antioxidant pathways,

Thank you for your advice.

Now, we cannot get materials smoothly. Therefore, we added a sentence in Discussion (Line: 451)

"In the future, we have to monitor the ROS production in cells treated with 11αOH-KA and A. lavenia extracts."

Reviewer 3 Report

The authors have made substantial positive changes to the manuscript.

Several minor edits suggested:

Line 324: choline and betaine are noted to be oxidative stress inducers, yet their levels are increased with 11aOH-KA and in the discussion you note that betaine confers protection against oxidative stress. Please clarify the known activities of choline/betaine as you interpret the higher levels in treated yeast.

Line 337: Table 2 footnote - "statistical analyses were performed..."

Figure 6B: Identify the grey and black bars.

Author Response

(Comment)

Line 324: choline and betaine are noted to be oxidative stress inducers, yet their levels are increased with 11aOH-KA and in the discussion you note that betaine confers protection against oxidative stress. Please clarify the known activities of choline/betaine as you interpret the higher levels in treated yeast.

Thank you for your advice.

We are sorry for misleading the Line324: choline and betaine are known as stress protectants. Therefore we corrected the sentence as

"On the contrary, betaine and choline (other stress protectants) were significantly increased by 7.5- and 1.85-fold, respectively, suggesting a cellular homeostasis alteration occurred following treatment with 11αOH-KA that involves a wide array of stress response mechanisms, which might lead to the lifespan extension of S. pombe

We have already cited some examples in Discussion Line 479-488 [41-45].

We added "It is intriguing to note that betaine has a principle role in the acquisition of stress tolerance in various organisms,.."

(Comment)

Line 337: Table 2 footnote - "statistical analyses were performed..."

Thank you very much for your suggestion.

We corrected the sentence.

(Comment)

Figure 6B: Identify the grey and black bars.

Thank you very much for your suggestion.

We added interpretation in the legend.

Thank you very much for your careful reading of our manuscript.